# Set2 histone methyltransferase regulates transcription coupled-nucleotide excision repair in yeast

Kathiresan Selvam[1], Dalton A. Plummer[1], Peng Mao[2], John J. Wyrick[1,3]*

1 School of Molecular Biosciences, Washington State University, Pullman, Washington, United States of America, 2 Department of Internal Medicine, Program in Cellular and Molecular Oncology, University of New Mexico Comprehensive Cancer Center, Albuquerque, New Mexico, United States of America, 3 Center for Reproductive Biology, Washington State University, Pullman, Washington, United States of America

* jwyrick@wsu.edu

**Data Availability Statement:** The CPD-seq data generated or analyzed in this study have been submitted to the NCBI Gene Expression Omnibus (GEO; https://www.ncbi.nlm.nih.gov/geo/) under

## Abstract

Helix-distorting DNA lesions, including ultraviolet (UV) light-induced damage, are repaired by the global genomic-nucleotide excision repair (GG-NER) and transcription coupled-nucleotide excision repair (TC-NER) pathways. Previous studies have shown that histone post-translational modifications (PTMs) such as histone acetylation and methylation can promote GG-NER in chromatin. Whether histone PTMs also regulate the repair of DNA lesions by the TC-NER pathway in transcribed DNA is unknown. Here, we report that histone H3 K36 methylation (H3K36me) by the Set2 histone methyltransferase in yeast regulates TC-NER. Mutations in Set2 or H3K36 result in UV sensitivity that is epistatic with Rad26, the primary TC-NER factor in yeast, and cause a defect in the repair of UV damage across the yeast genome. We further show that mutations in Set2 or H3K36 in a GG-NER deficient strain (i.e., rad16Δ) partially rescue its UV sensitivity. Our data indicate that deletion of *SET2* rescues UV sensitivity in a GG-NER deficient strain by activating cryptic antisense transcription, so that the non-transcribed strand (NTS) of yeast genes is repaired by TC-NER. These findings indicate that Set2 methylation of H3K36 establishes transcriptional asymmetry in repair by promoting canonical TC-NER of the transcribed strand (TS) and suppressing cryptic TC-NER of the NTS.

## Author summary

Exposure to UV light causes damage to DNA, which must be efficiently repaired to avoid cell death or carcinogenesis. UV damage is repaired by the nucleotide excision repair pathway, which is triggered either by directly sensing UV damage (known as global genomic repair) or by sensing the stalling of an RNA polymerase enzyme at UV damage during transcription (known as transcription coupled repair). Repair of UV damage must take place in the context of DNA tightly packaged with histone proteins into chromatin. Previous studies have shown that global genomic repair can be facilitated by epigenetic modifications of histone proteins in chromatin. However, whether transcription coupled repair is also regulated by these or other histone modifications was previously unclear. Here, we

accession numbers GSE179794, GSE149082, GSE131101, and GSE194101.

**Funding:** This study was supported by National Institute of Environmental Health Sciences grants R01ES028698 (J.J.W.), R21ES029302 (J.J.W. and P.M.), and R01ES032814 (J.J.W.). The funders had no role in study design, data collection and analysis, decision to publish, or preparation of the manuscript.

**Competing interests:** The authors have declared that no competing interests exist.

show that a specific histone modification catalyzed by the Set2 protein in yeast promotes transcription coupled repair. Surprisingly, in the absence of Set2, a new form of transcription coupled repair is activated at sites of cryptic transcription, which can restore UV resistance to otherwise repair-deficient cells. Since Set2 homologs are frequently mutated in human cancers, these findings have potentially important implications for carcinogenesis and chemotherapeutic resistance.

## Introduction

The nucleotide excision repair (NER) pathway is critically important for removing bulky, helix-distorting DNA lesions, such as UV-induced cyclobutane pyrimidine dimers (CPDs) [1,2]. NER is comprised of two subpathways: global genomic-NER (GG-NER), which repairs DNA lesions throughout the genome [3], and transcription coupled-NER (TC-NER), which specifically repairs DNA lesions on the transcribed strand (TS) of actively transcribed genes [4,5]. These two subpathways primarily differ in the means of lesion recognition. GG-NER directly senses the helix distortion induced by bulky adducts via the DNA damage sensors XPC and UV-DDB in mammalian cells and Rad4 and Rad7/Rad16/Elc1 in *Saccharomyces cerevisiae* [3,6,7]. In contrast, TC-NER recognizes RNA polymerase II (Pol II) stalling at a bulky, helix-distorting lesion, which triggers repair via the Cockayne syndrome B (CSB) gene in human cells, and its homolog Rad26 in *S. cerevisiae* [4,5,8]. How CSB or Rad26 is specifically recruited to stalled Pol II is unknown.

Both NER pathways must repair lesions in the context of DNA packaged with histones into nucleosomes and higher order chromatin structures. Chromatin limits the accessibility of DNA lesions to repair factors, but can also serve as a signaling platform to promote repair [9–11]. For example, histone acetylation by the Gcn5 and NuA4 histone acetyltransferases has been shown to promote GG-NER of UV damage in yeast [12–14]. Histone H3K79 methylation by the Dot1 histone methyltransferase also promotes GG-NER [15–17]. However, whether histone PTMs regulate TC-NER is unknown.

Actively elongating Pol II recruits the Set2 histone methyltransferase, which catalyzes all three states of methylation (mono-, di-, and tri-) of histone H3 K36 [18]. One of the primary functions of H3K36 methylation is to prevent spurious sense or antisense transcription from cryptic promoters inside of genes by recruiting and activating the Rpd3S histone deacetylase complex [19,20]. H3K36 methylation also regulates RNA splicing [21,22] and DNA double strand break repair [23,24]. Previous studies have shown that H3K36 methylation by Set2 plays a role in recruiting the TC-NER factor Rad26 to transcribed yeast genes in undamaged yeast cells [25], where it presumably functions to promote Pol II elongation. However, to what extent H3K36 methylation regulates Rad26-dependent TC-NER is unclear.

Here, we report that mutations in Set2 or H3K36 result in UV sensitivity that is linked to the Rad26-dependent TC-NER pathway. We further show using our genome-wide CPD-seq method [26] that Set2 is required for efficient repair of both the NTS and TS of yeast genes. However, in the absence of GG-NER (*rad16Δ*), deletion of *SET2* increases resistance to UV, which we show is due to TC-NER associated with cryptic antisense transcription along the NTS of yeast genes.

## Results

### Deletion of *SET2* causes UV sensitivity linked to a defect in TC-NER

To determine whether Set2 plays role in the repair of UV damage, we analyzed the UV sensitivity of a yeast strain lacking *SET2* (*set2Δ*). A spotting assay revealed that *set2Δ* cells exhibited

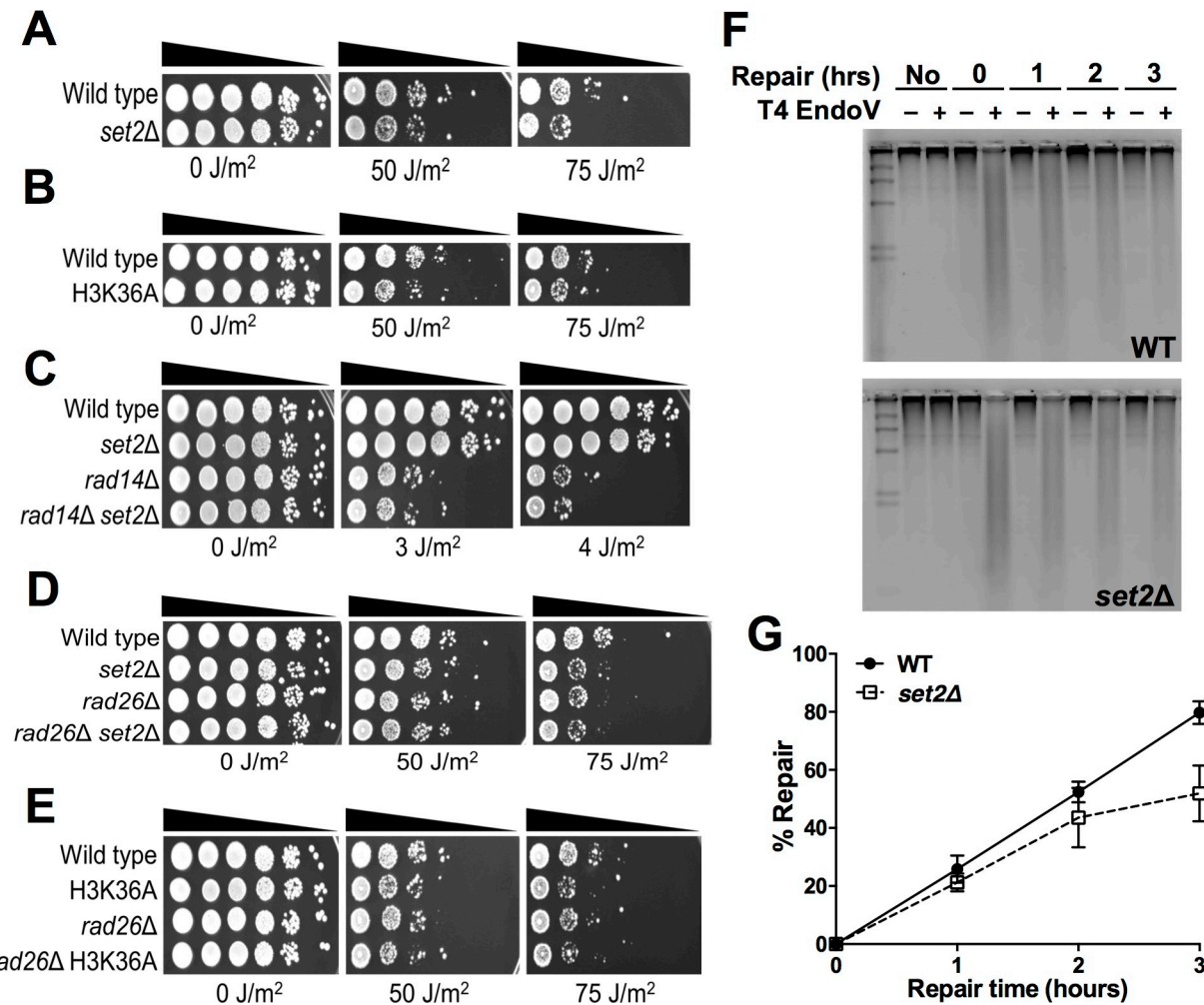

**Fig 1. Mutation of *SET2* or H3K36 causes UV sensitivity that is epistatic with *RAD26*. (A-E)** Indicated mutant yeast strains were serially 10-fold diluted, spotted on YPD plates, and exposed to the specified doses of UVC light. **(F)** Representative alkaline gel of bulk repair of UV-induced CPD lesions in WT and *set2Δ* mutant yeast cells. Genomic DNA was isolated at the indicated time following damage induction with 100 J/m² UVC light and treated with or without (+/−) T4 endonuclease V and run on the alkaline gel electrophoresis. **(G)** Quantification of CPD repair in WT and *set2Δ* cells based on alkaline gel analysis plotted as the mean ± SEM of three replicate experiments. Source data is in S1 Data.

increased UV sensitivity relative to wild-type (WT), especially at higher doses (Fig 1A). In yeast, Set2 is solely responsible for methylation of H3K36 [18], so we tested whether the loss of H3K36me also causes UV sensitivity. Analysis of an H3K36A point mutation, which also eliminates H3K36me (S1A Fig), showed a similar increase in UV sensitivity (Fig 1B). This result is consistent with previous reports [27–29] and indicates that H3K36 methylation by Set2 is important for UV resistance.

To determine whether the observed increase in UV sensitivity is due to a defect in NER, we performed epistasis analysis in which we tested the UV sensitivity of *set2Δ* in combination with a deletion in *RAD14*, which is required for both GG-NER and TC-NER pathways [7]. Deletion of *SET2* in the *rad14Δ* mutant background did not further enhance its UV sensitivity (Fig 1C), consistent with the hypothesis that Set2 functions in Rad14-mediated NER. *RAD14* is the only NER gene that contains an intron in yeast, and it has been recently reported that *SET2* deletion affects the splicing of a subset of *RAD14* transcripts [22]. To test whether a potential defect in *RAD14* splicing is responsible for UV sensitivity in the *set2Δ* mutant, we

used CRISPR to delete the *RAD14* intron (*RAD14*-intronΔ) in the *set2*Δ strain, in order to eliminate any possible splicing defect. The *set2*Δ mutant in the *RAD14*-intronΔ background showed a similar UV sensitivity as the *set2*Δ single mutant (S1B Fig), indicating that *SET2* deletion still causes the same level of UV sensitivity even when *RAD14* splicing is not required. Moreover, analysis of published gene expression data sets [30–34] indicated that no NER genes showed a significant decrease in expression level in the *set2*Δ mutant background. Taken together, these findings suggest that Set2 directly promotes NER.

To determine which NER pathway is regulated by Set2, we performed epistasis analysis by constructing double mutants of *set2*Δ with *RAD26*, which is specifically required for TC-NER [8], and *RAD16*, which is specifically required for GG-NER [6]. While *set2*Δ is not epistatic with *rad16*Δ, but actually rescues its UV sensitivity (see below), deletion of *SET2* in *rad26*Δ cells did not further enhance the modest UV sensitivity of the *rad26*Δ single mutant (Fig 1D). Similarly, the *rad26*Δ H3K36A double mutant was not more UV sensitive than either single mutant (Fig 1E), indicating that Set2 methylation of H3K36 is epistatic with Rad26, and therefore may function in the Rad26-dependent TC-NER pathway.

## Set2 is important for efficient TC-NER

Since epistasis analysis indicated that Set2 plays a role in NER, we investigated whether Set2 is required for efficient NER of UV damage. To test whether Set2 is generally required for bulk repair of CPD lesions, we performed a T4 endonuclease V digestion and alkaline gel electrophoresis assay [35,36] to measure global repair of CPD lesions in UV-irradiated yeast (Fig 1F). The data revealed that the *set2*Δ did not significantly affect overall repair of CPD lesions compared to the WT control (Fig 1G), although there was a marginal repair defect at the 3 hour (hr) time point ($P = 0.055$). While these data suggest that Set2 plays a marginal role in global NER, they do not address whether Set2 specifically promotes TC-NER, since the TC-NER pathway only repairs actively transcribed genes and plays a relatively minor role in overall repair in yeast [8].

To test whether Set2 plays a role in TC-NER, we used the CPD-seq method [26] to map CPD lesions immediately after UV irradiation (0hr) and following 2 hours of repair (2hr) in both WT and *set2*Δ strains. The CPD-seq method couples T4 endonuclease V and ApeI digestion to next generation sequencing to map UV-induced CPD lesions at single nucleotide resolution and in a strand-specific manner across the yeast genome [26]. We analyzed the fraction of CPD lesions remaining (i.e., unrepaired) after 2hr repair relative to 0hr time point for different bins along both the non-transcribed strand (NTS) and transcribed strand (TS) of ~5000 yeast genes with previously defined transcription start site (TSS) and transcription end sites (TES; based on mapped polyadenylation sites [37], see Materials and Methods). Because there can be some variability in the absolute levels of CPDs measured using CPD-seq [26], we normalized the fraction of CPDs remaining using the alkaline gel assays (Fig 1F and 1G), following our published protocol [38].

In WT cells, following 2hr of repair there were fewer unrepaired CPDs along the TS relative to the NTS or adjacent intergenic DNA (Fig 2A). This result is consistent with prior reports [26,39,40] and represents more rapid repair of the TS by the TC-NER pathway. In the *set2*Δ mutant, there were more unrepaired CPDs on both DNA strands, particularly along the TS (Fig 2B), indicating that Set2 may regulate both GG-NER and TC-NER pathways. The repair defect along the TS was not to the same extent as observed for a *rad26*Δ mutant [40], suggesting that Set2 is not essential for TC-NER, but may instead be required for efficient TC-NER in chromatin.

To measure the impact of Set2 on TC-NER, we analyzed the log ratio of unrepaired CPDs on the TS relative to the NTS across ~5000 yeast genes. This method quantifies the magnitude

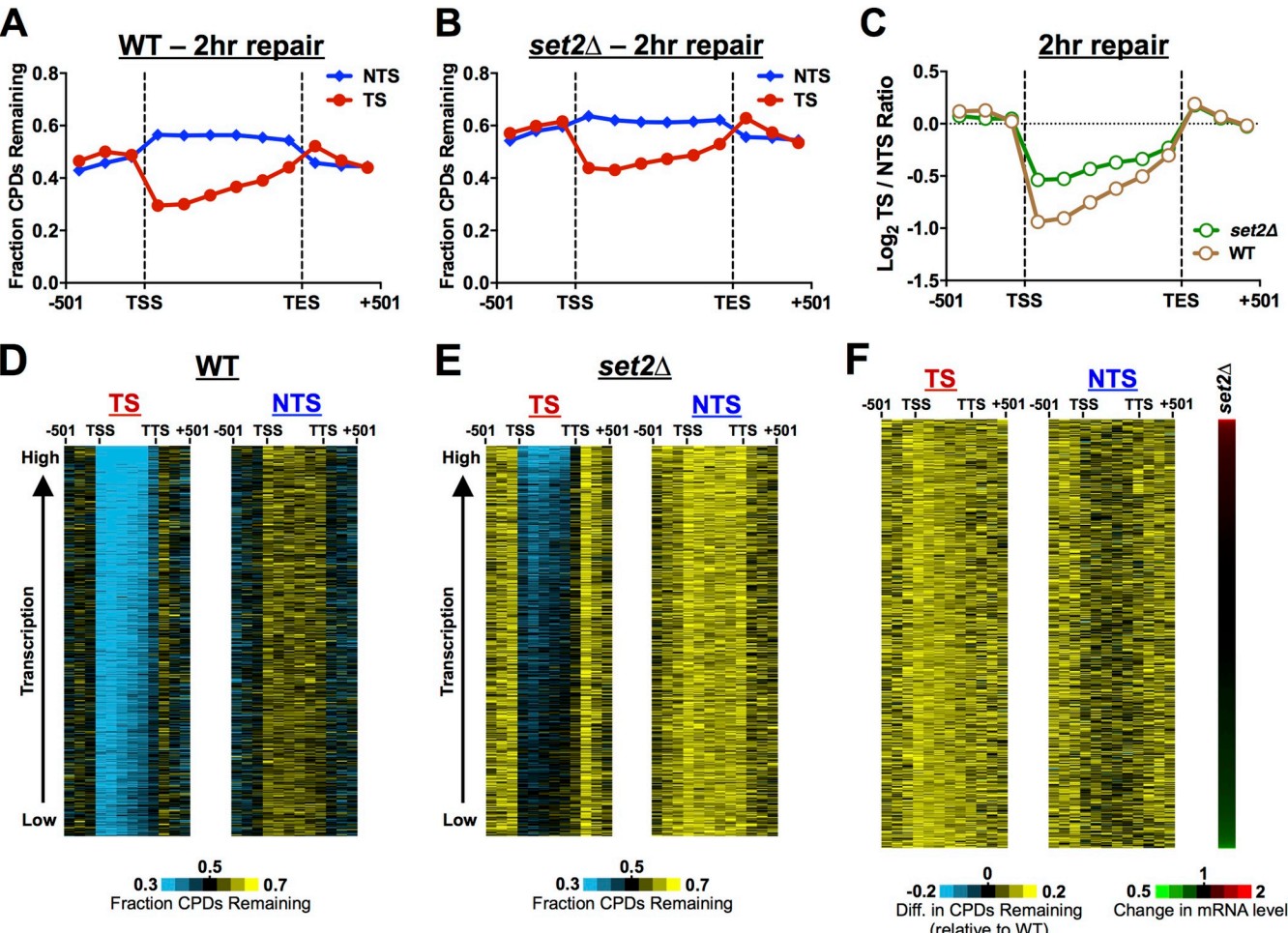

**Fig 2. CPD-seq analysis of genome-wide repair of *set2Δ* mutant cells reveals a defect in TC-NER. (A)** The average fraction of CPDs remaining after 2 hours (2hr) of repair was plotted from the CPD-seq data for both the transcribed strand (TS) and non-transcribed strand (NTS) for approximately 5000 yeast genes in WT cells. Number of unrepaired CPDs after 2hr was normalized to the initial CPD levels (0hr) to calculate the fraction of CPDs remaining. CPD-seq data for each gene was divided into 6 equally sized bins and the fraction of CPDs remaining was analyzed in each bin. The fraction of CPDs remaining was also analyzed for 3 bins each consisting of 167 bp of flanking DNA upstream of transcription start site (TSS) or downstream of transcription end site (TES) for each gene. CPD-seq data was scaled using the overall repair data for the 2hr time point from alkaline gel analysis (Fig 1G). **(B)** Same as panel **A**, except for *set2Δ* cells. **(C)** The log$_2$ ratio of unrepaired CPDs on the TS relative to the NTS in the WT and *set2Δ* cells is shown to quantify the TC-NER defect in *set2Δ* cells. Less efficient repair of the TS by TC-NER in *set2Δ* cells is indicated by the log$_2$ TS/NTS being closer to zero. **(D,E)** Gene plot analysis of WT and *set2Δ* cells to examine CPD repair within genes throughout the yeast genome. The fraction of CPDs remaining following 2hr of repair (relative to 0hr) is plotted for 6 equally sized bins comprising the transcribed region of each gene, for both the TS and NTS. 3 bins consisting of 167 bp regions upstream of TSS and 3 bins downstream of the TES are also depicted. Genes were ordered based on transcription frequency [42]. **(F)** Same as panels **D,E**, except showing difference in the fraction of unrepaired CPDs at the 2hr time point in *set2Δ* mutant cells relative to WT. Genes are ordered by their fold-change in mRNA levels in the *set2Δ* mutant (right panel). mRNA expression data for *set2Δ* is from [32–34].

of the repair asymmetry between the TS and NTS of yeast genes due to TC-NER, and therefore can be used to characterize changes in TC-NER efficiency, even if GG-NER is also altered in the mutant cells [41]. In WT cells, the log$_2$ TS/NTS ratio was much less than zero in the transcribed DNA (i.e., region between TSS and TES; see Fig 2C), due to faster repair of the TS by TC-NER. However, in the *set2Δ* mutant, the log$_2$ TS/NTS ratio was closer to zero (Fig 2C), likely reflecting impaired TC-NER of the TS. To confirm these findings, we repeated the CPD-seq experiments for *set2Δ* and WT cells, both at 2hr and 3hr following UV irradiation, and normalized the CPD-seq data using the alkaline gel data for each time point, as described above. Using this normalization, we observed consistent repair defects at both the 2hr and 3hr

time points, especially along the TS (S2A–S2D Fig). Notably, the $\log_2$ TS/NTS ratio was closer to zero than the WT control at both the 2hr and 3hr time points (S2E and S2F Fig), consistent with a defect in TC-NER of the TS in the *set2Δ* mutant. These results confirm that Set2 is important for efficient repair in yeast, particularly of the TS by the TC-NER pathway.

To test whether Set2 is required for repair of a specific subset of genes or throughout the genome, we analyzed CPD repair along the TS and NTS of individual genes throughout the genome. The fraction of unrepaired CPDs along the TS and NTS was visualized for yeast genes ordered by their transcription frequency [42]. In WT cells, there were fewer unrepaired CPDs along the TS relative to the NTS (Figs 2D and S2G), particularly in highly expressed genes, but also in those that are lowly expressed, consistent with previous results [40]. In the *set2Δ* mutant, there were more unrepaired CPDs along the TS (and NTS) for most genes (Figs 2E and S2H), suggesting that Set2 is required for TC-NER at genes throughout the genome, regardless of transcription frequency. There was also apparent slower repair of flanking DNA (i.e., upstream of the TSS or downstream of the TTS) in the *set2Δ* mutant (Fig 2A, 2B, 2D and 2E). This could reflect slower GG-NER or a defect in TC-NER of neighboring genes.

Set2 is known to regulate the expression of a number of yeast genes, so we examined whether changes in Set2-regulated gene expression might potentially contribute to the observed repair defects. We analyzed the difference in the fraction of unrepaired CPDs in the *set2Δ* mutant relative to WT following 2hr of repair for all yeast genes, in this case ordered by their change in expression in a *set2Δ* mutant [32–34]. There were more unrepaired CPDs in the *set2Δ* mutant relative to WT across nearly all genes, particularly along the TS, regardless of expression changes in the *set2Δ* mutant (Fig 2F). These results indicate that Set2 regulates the repair of yeast genes independent of any changes in gene expression.

To analyze repair of the TS at single nucleotide resolution, we plotted the fraction of CPD lesions remaining around the transcription start site (TSS) of ~5200 yeast genes (Fig 3). In WT cells, there were fewer unrepaired CPDs along the TS, beginning immediately after the TSS, while more unrepaired CPDs were present along the NTS, particularly at nucleosomes (Fig 3A). Slower repair of CPD lesions in nucleosomal DNA relative to adjacent linker DNA (Fig 3A) is consistent with our previous findings [26,39,41]. Comparing the WT and *set2Δ* mutant data revealed more unrepaired CPDs in the *set2Δ* mutant throughout the TS (Fig 3B), consistent with the model that Set2 promotes efficient TC-NER. While TC-NER of most of the TS is dependent upon Rad26, repair of TS within the TSS-proximal half of the +1 nucleosome is largely independent of Rad26 [40]. In the *set2Δ* mutant, there were also more unrepaired CPDs in the TS within this TSS-proximal half of the +1 nucleosome (Fig 3C and 3D), indicating that Set2 may also facilitate Rad26-independent repair. Finally, repair of the NTS appeared somewhat decreased and peaks of unrepaired CPDs at nucleosomes in the NTS are less prominent in the *set2Δ* mutant relative to WT (Fig 3A and 3B). This latter observation may be a consequence of increased histone exchange and dynamics in the coding regions of genes, which has been previously reported for *set2Δ* mutant cells [43].

## Deletion of *SET2* rescues the UV sensitivity of a GG-NER deficient mutant

While yeast mutants that affect TC-NER normally have elevated UV sensitivity in a GG-NER deficient background (e.g., *rad16Δ* or *rad7Δ* mutant [44,45]), our data revealed that deletion of *SET2* partially rescued the UV sensitivity of a *rad16Δ* mutant (Fig 4A). The H3K36A mutant also rescued the *rad16Δ* UV sensitivity (Fig 4B), indicating that loss of H3K36 methylation partially restores UV resistance to GG-NER deficient yeast cells. Set2 methylation normally functions to suppress cryptic transcription, which can be sense or antisense to normal transcription [19,20,46,47]. We hypothesized that activation of cryptic antisense transcription

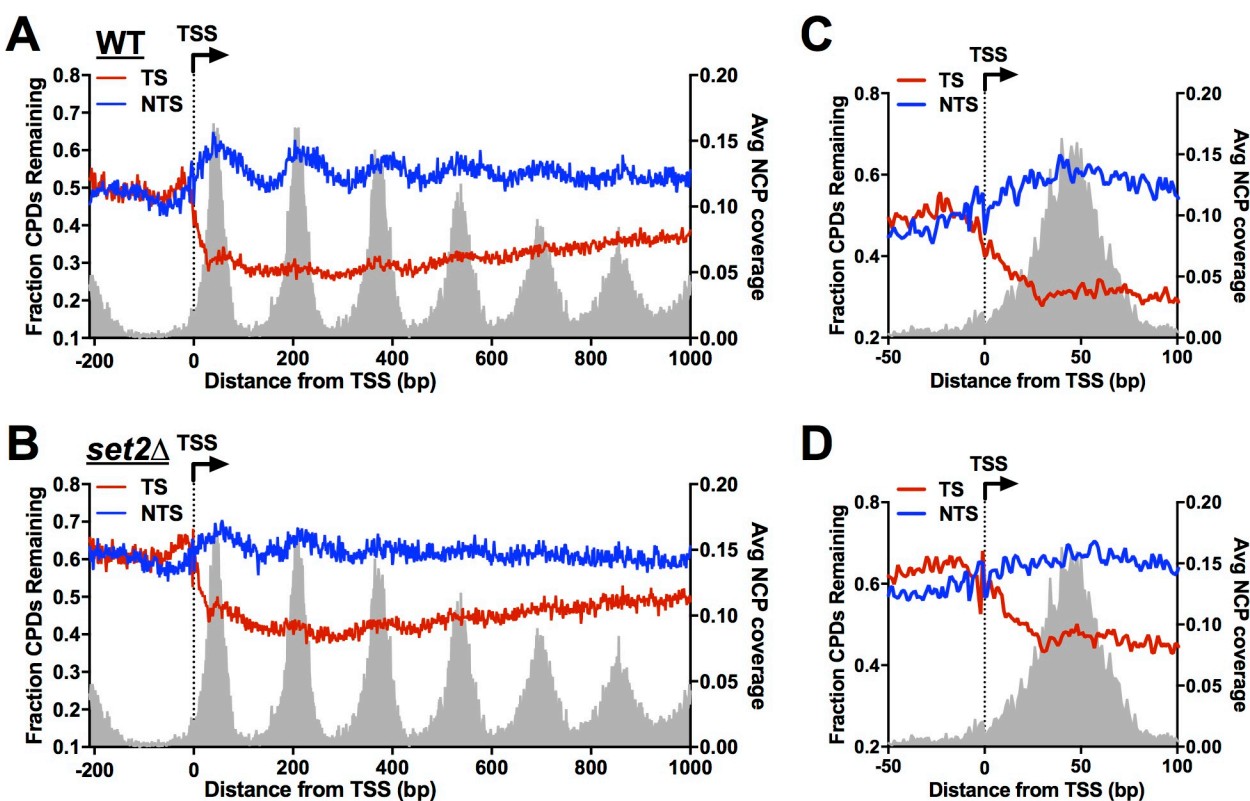

**Fig 3. High-resolution CPD-seq analysis of repair around the transcription start site in *set2Δ* mutant cells.** (**A**) Single nucleotide resolution analysis of CPD-seq data on both transcribed strand (TS) and nontranscribed strand (NTS) of ~5200 yeast genes around the transcription start site (TSS). The plots depict fraction of unrepaired CPDs following 2hr repair relative to no repair (0hr) for both TS and NTS in WT cells. The data is plotted for each nucleotide positioning spanning -200 bp upstream and +1000 bp downstream of the TSS. CPD-seq data was scaled using the overall repair data for the 2hr time point from alkaline gel analysis (Fig 1G). The positions of nucleosome dyads [77] are shown for reference. (**B**) Same as panel **A**, except the data is from *set2Δ* cells. (**C,D**) Close-up of CPD-seq data near the TSS within the region of +1 nucleosome in the WT (**C**) and *set2Δ* (**D**) cells. Nucleosome positioning data for the +1 nucleosome is shown for reference.

in a *set2Δ* (or H3K36A) mutant would stimulate repair of the NTS through the TC-NER pathway, thereby restoring UV resistance. Deletion of the TC-NER factor *RAD26* in a *rad16Δ* background enhances UV sensitivity (Fig 4C), as expected [44]. Importantly, the *set2Δ* mutant is unable to rescue the UV sensitivity of a *rad16Δ* mutant if the TC-NER factor *RAD26* is also deleted (Fig 4C), consistent with our hypothesis.

This hypothesis predicts that deletion of *SET2* should stimulate repair of the NTS in a *rad16Δ* mutant background. To test this prediction, we used CPD-seq to measure genome-wide repair of CPD lesions in a *rad16Δ set2Δ* double mutant at 0hr and 2hr following UV irradiation, and compared these data to our published *rad16Δ* CPD-seq data set [39]. In both the *rad16Δ* and *rad16Δset2Δ* mutant strains, there were more unrepaired CPD lesions along the NTS and the periodic GG-NER pattern in nucleosomes was absent (S3 Fig), consistent with our previous findings that GG-NER efficiency is modulated in nucleosomal DNA [39]. Deletion of *SET2* in the *rad16Δ* background resulted in more unrepaired CPDs along the TS (Fig 4D), confirming that Set2 promotes TC-NER along the TS. However, there were fewer unrepaired CPDs along the NTS in the *rad16Δset2Δ* double mutant relative to the *rad16Δ* single mutant (Fig 4D), indicating that Set2 normally inhibits repair of the NTS and deletion of *SET2* somehow activates its repair in the GG-NER deficient *rad16Δ* strain. There was no difference in CPDs remaining in DNA flanking the TSS or TES (Fig 4D), suggesting that Set2 primarily

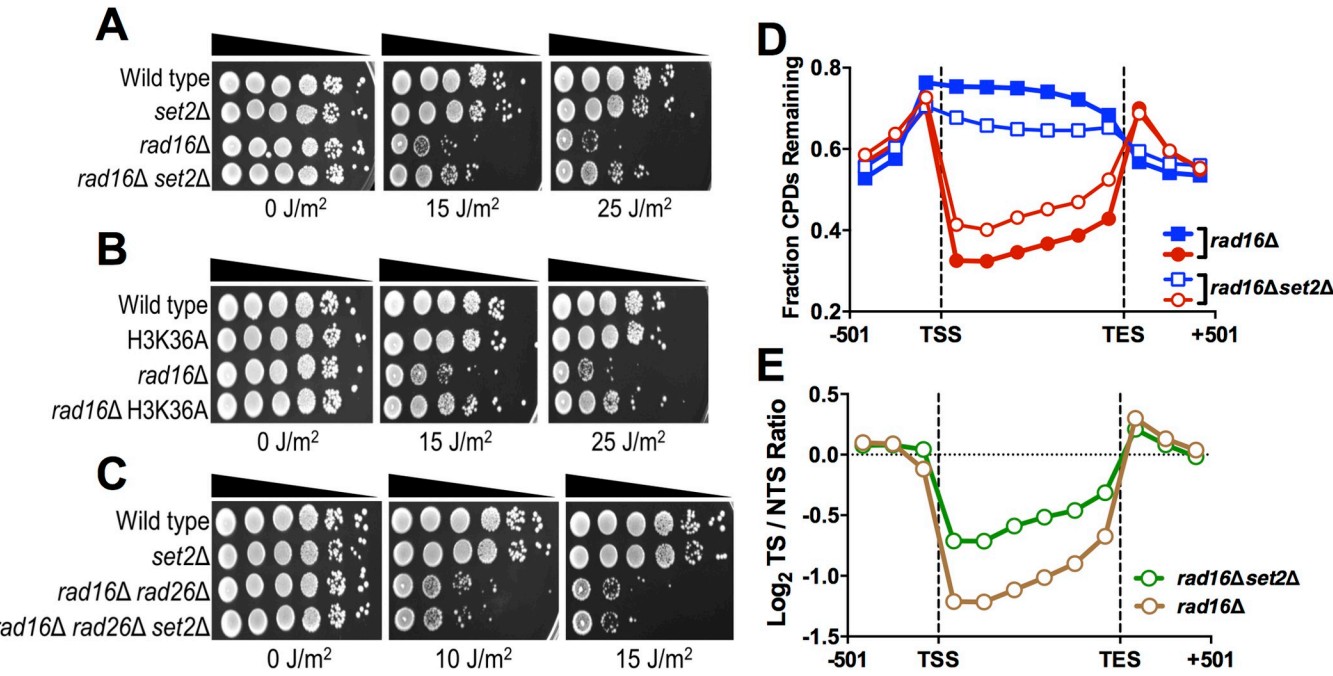

**Fig 4. Deletion of *SET2* restores UV resistance in a GG-NER deficient strain by promoting repair of the NTS. (A-C)** Indicated mutant yeast strains were serially 10-fold diluted, spotted, and exposed to the specified UVC doses. **(D)** The average fraction of CPDs remaining after 2-hour repair (relative to 0hr control) was plotted from CPD-seq data for both the transcribed strand (TS) and nontranscribed strand (NTS) for approximately 5000 yeast genes in *rad16Δ* and *rad16Δset2Δ* cells. CPD-seq data for each gene was divided into 6 equally sized bins and fraction of CPDs remaining was analyzed in each bin. The fraction of CPDs remaining in 3 bins consisting of consecutive 167 bp regions of flanking DNA upstream of transcription start site (TSS) or downstream of transcription end site (TES) is also depicted. Red symbols/line indicates the transcribed strand (TS); blue symbols/line indicates the non-transcribed strand (NTS). **(E)** The log ratio of unrepaired CPDs on the TS relative to the NTS in the *rad16Δ* and *rad16Δset2Δ* cells is depicted.

regulates repair within gene coding regions, not in intergenic DNA. The *rad16Δset2Δ* strain showed much less transcriptional asymmetry in repair (i.e., log$_2$ TS/NTS ratio) than the *rad16Δ* control (Fig 4E), due to both slower repair on the TS and faster repair of the NTS. A repeat of the *rad16Δset2Δ* experiment, which was scaled so that the fraction of CPDs remaining in flanking intergenic regions matched that of *rad16Δ* (see Materials and Methods), showed similar results (S4 Fig). Taken together these findings indicate that deletion of *SET2* restores UV resistance to a GG-NER defect strain (i.e., *rad16Δ*) by partially restoring repair of the NTS of yeast genes.

## Loss of *SET2* activates cryptic TC-NER associated with antisense transcription

Deletion of *SET2* results in the accumulation of Set2-repressed antisense transcripts (SRATs) due to activation of cryptic antisense transcription of the NTS [19,46,47]. We tested whether cryptic antisense transcription of SRATs is associated with faster repair of the NTS of the associated yeast genes in the *rad16Δset2Δ* mutant. Analysis of 463 polyA SRAT-associated genes showed fewer unrepaired CPDs along the NTS in the *rad16Δset2Δ* mutant (relative to the *rad16Δ* control; see Fig 5A) compared to the rest of the genome (Fig 5B). Quantification of these data confirmed that there were relatively fewer unrepaired CPDs along the NTS of polyA SRAT-associated genes compared to the rest of the genome (S5A Fig). Similar results were obtained for the replicate *rad16Δset2Δ* experiment (S5B Fig). A similar analysis of repair of all SRAT-associated genes yielded similar results (S6 Fig). While repair of the NTS was faster in

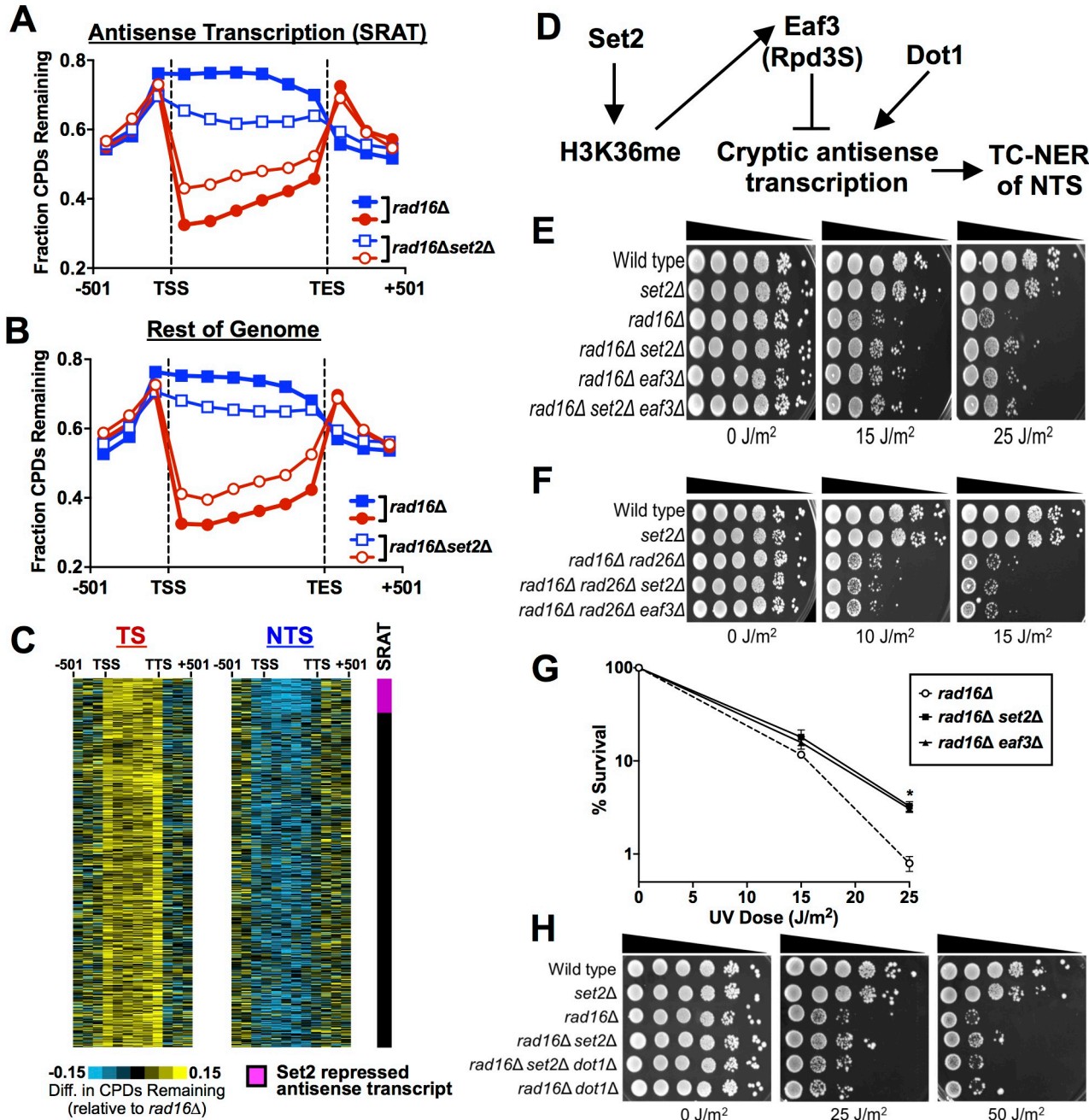

**Fig 5. Loss of *SET2* induces repair of the NTS particularly at genes with cryptic Set2-repressed antisense transcripts (SRATs). (A)** The average fraction of CPDs remaining after 2hr of repair (relative to 0hr) was plotted for CPD-seq data for 463 genes with polyadenylated SRATs in *rad16Δ* and *rad16Δset2Δ* cells. List of polyA SRAT genes is from [46]. Red symbols/line indicates the transcribed strand (TS); blue symbols/line indicates the non-transcribed strand (NTS). **(B)** Same as panel **A**, except CPD-seq data is shown for all non-polyA-SRAT genes. **(C)** Gene plot analysis showing difference in the fraction of unrepaired CPDs following 2hr of repair along the TS (left panel) and NTS (right panel) of yeast genes in *rad16Δset2Δ* cells relative to *rad16Δ* cells. PolyA SRAT genes are sorted at the top of the gene plot. **(D)** Model of how loss of Set2 (or Eaf3) promotes TC-NER of the NTS by activating cryptic antisense transcription. Dot1 is required to activate cryptic transcription for many genes in a *set2Δ* mutant [50]. **(E,F)** The indicated mutant yeast strains were serially 10-fold diluted, spotted, and exposed to specified UVC doses **(G)** Quantitative UV survival assay of the indicated yeast cells. The graph represents the quantification of UV survival from at least three independent experiments ±SEM. *P≤0.05. Source data is in S2 Data. **(H)** UV sensitivity assay (similar to panels **E,F**) showing that Dot1, which is required for activation of cryptic transcription in a *set2Δ* mutant (see panel **D**), is also required for rescue of UV sensitivity in a *rad16Δset2Δ* mutant.

SRAT-associated genes (Figs 5A and S6A), the NTS of non-SRAT genes (i.e., the rest of the genome) also had fewer unrepaired CPDs in the *rad16Δset2Δ* mutant relative to the *rad16Δ* control (Figs 5B and S6B). Gene plot analysis confirmed that unrepaired CPDs were reduced along the NTS in the *rad16Δset2Δ* mutant relative to the *rad16Δ* control in genes across the yeast genome (Fig 5C), but this was especially apparent at polyA SRAT-associated genes (see top of Fig 5C). These results indicate that loss of Set2 can generally promote repair on the NTS when GG-NER is deficient. In contrast, there was essentially the same repair defect in the *rad16Δset2Δ* mutant along the TS in SRAT genes and the rest of the genome (Figs 5A, 5B, 5C, S6A, and S6B), indicating that the TC-NER defect on the TS is likely not due to antisense transcription in *set2Δ* cells. Taken together, these analyses suggest that Set2 normally suppresses repair of the NTS by the TC-NER pathway, but this cryptic TC-NER pathway is activated by deletion of *SET2* due to the induction of antisense transcription, thereby restoring UV resistance in GG-NER deficient cells.

Our model suggests that ongoing Pol II transcription of the NTS in *set2Δ* mutant cells promotes its repair via the TC-NER pathway. However, an alternative possibility is that the resulting cryptic antisense RNAs themselves somehow promote repair of the NTS. Many of these cryptic antisense RNAs are rapidly degraded by either the exonuclease Xrn1 or the exosome [46]. To test this possibility, we measured whether the UV sensitivity in *set2Δ* mutant yeast is modulated by stimulating the accumulation of SRATs RNAs, either by deleting *XRN1* or the *RRP6* component of the exosome. Deletion of *XRN1* resulted in slightly elevated UV resistance in a WT background (S7A Fig), but the double mutant with *set2Δ* did not modulate the UV sensitivity of a *set2Δ* mutant strain (S7A Fig) nor the UV resistance imparted by *set2Δ* in a *rad16Δ* mutant background (S7B Fig). Similarly, deletion of *RRP6* did not modulate the UV sensitivity of the *set2Δ* mutant or UV resistance of the *rad16Δset2Δ* double mutant (S7C and S7D Fig). Additionally, deletion of *XRN1* or *RRP6* did not rescue the UV sensitivity of a *rad16Δ* mutant (S7B and S7D Fig). These findings suggest that the UV resistance imparted by deletion of *SET2* in a *rad16Δ* mutant background is likely not mediated by the antisense transcripts themselves.

Histone H3K36 methylation by Set2 suppresses cryptic antisense transcription by recruiting the Rpd3S histone deacetylase complex via its Eaf3 subunit [19,48]. To further verify our hypothesis that TC-NER associated with Set2-repressed antisense transcripts promotes repair of the NTS (Fig 5D), we tested the effects of deleting *EAF3* in the *rad16Δ* mutant background. Deletion of *EAF3* in the *rad16Δ* background partially rescued its UV sensitivity, similar to *set2Δ* (Fig 5E). Moreover, deletion of *EAF3* in *rad16Δ rad26Δ* mutant did not impart UV resistance (Fig 5F), confirming that Rad26-dependent TC-NER is required to restore the UV resistance of the *rad16Δ* mutant. Quantitative UV sensitivity assays revealed that both *set2Δ* and *eaf3Δ* rescued the *rad16Δ* UV sensitivity to a similar extent (Fig 5G). Set2 methylation can also recruit the Isw1b complex via its Ioc4 subunit [49]. However, deletion of *IOC4* did not modulate UV sensitivity in WT or *set2Δ* mutant cells, nor impart UV resistance in a *rad16Δ* mutant background (S7E and S7F Fig). Taken together, these results indicate that Eaf3 (and not Ioc4) is required for Set2-mediated suppression of cryptic TC-NER.

A recent study has shown that activation of cryptic transcription in *set2Δ* cells requires Dot1 [50]. To confirm that cryptic transcription is required to restore UV resistance in the *rad16Δset2Δ* strain, we tested the UV sensitivity of the *rad16Δset2Δ* mutant in a *dot1Δ* background. In a *dot1Δ* background, deletion of *SET2* did not restore UV resistance to a *rad16Δ* mutant (Fig 5H), indicating that cryptic transcription mediated by Dot1 is important for UV resistance. Taken together, these findings support our model that loss of Set2 and Eaf3-dependent repression of antisense transcription restores UV resistance to a GG-NER deficient strain by promoting cryptic TC-NER of the NTS (Fig 5D).

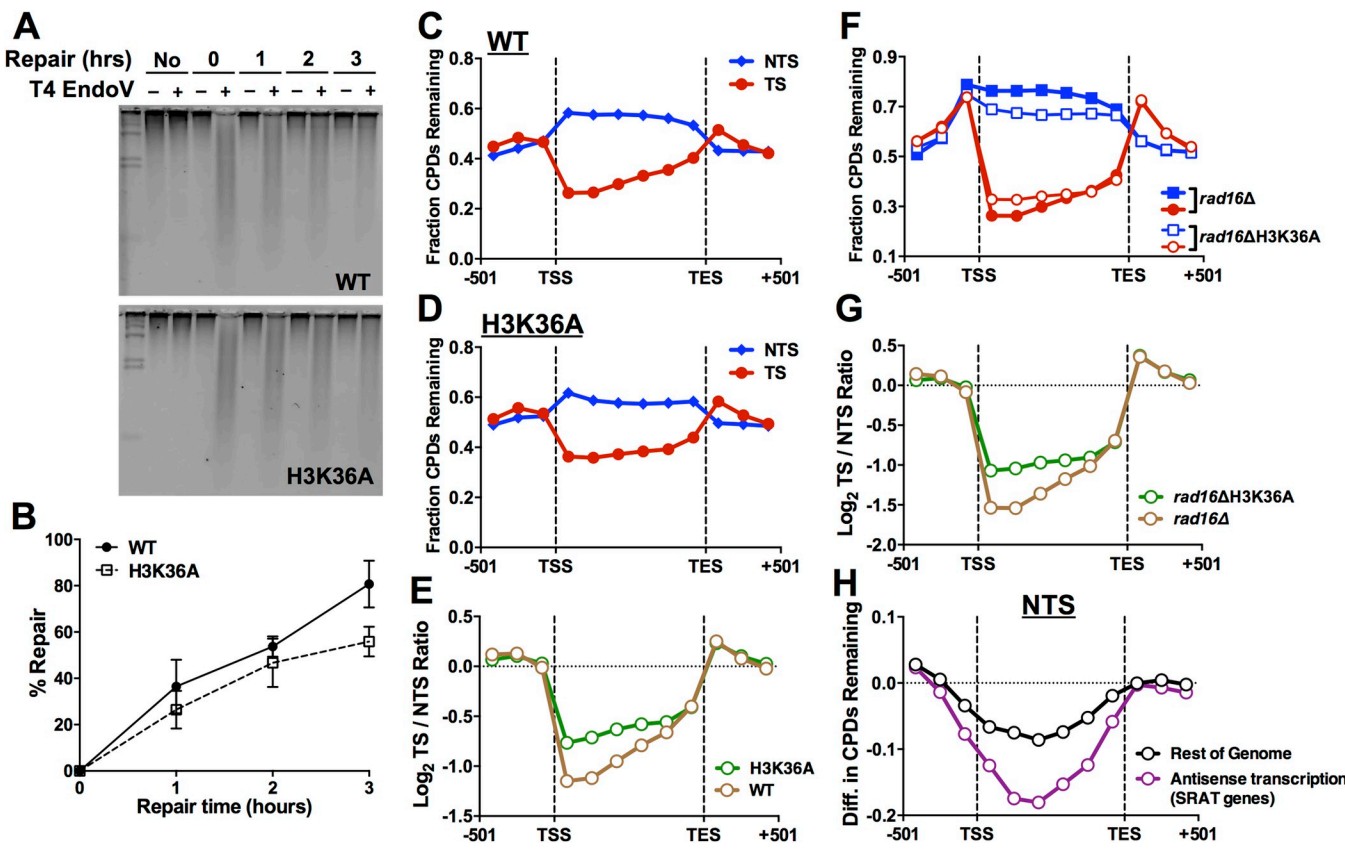

**Fig 6. Histone residue H3K36 regulates NER in a similar manner as Set2.** (**A**) Representative alkaline gel measuring overall repair of UV-induced CPD lesions in histone WT and histone H3K36A mutant yeast cells. (**B**) Quantification of alkaline gel analysis of overall repair of CPD lesions in yeast over a 3hr time course. Repair is plotted as the mean ± SEM of three replicate experiments. Source data is in S1 Data. (**C-D**) Bin plot analysis of CPD-seq repair data for histone WT (**C**) and H3K36A mutant (**D**) cells. The fraction of CPDs remaining after 2hr of repair relative to the 0hr time point is plotted for 6 equally-sized bins in the coding regions of ~5000 yeast genes, as well as three flanking bins (167 bp in length) upstream of the transcription start site (TSS) and downstream of the transcription end site (TES). CPD-seq data was scaled using the overall repair data for the 2hr time point from alkaline gel analysis (panel **B**). (**E**) Quantification of repair asymmetry, based on the log₂ ratio of fraction of CPDs remaining for the TS of yeast genes relative to the NTS for each bin (see panels **C,D** above) for histone WT and H3K36A mutant CPD-seq data. (**F**) Bin analysis of CPD-seq data for histone WT in a *rad16Δ* mutant background relative to the H3K36A *rad16Δ* double mutant. Fraction of CPDs remaining following 2hr of repair relative to 0hr is plotted for the NTS (blue) and TS (red) of yeast genes. The double mutant CPD-seq data was scaled so that the median ratio for CPDs remaining in the flanking (i.e., intergenic) bins in the double mutant relative to the *rad16Δ* single mutant was set to 1. (**G**) Same as panel **E** above, except repair asymmetry between the TS and NTS of yeast genes was determined for the *rad16Δ* single mutant and H3K36A *rad16Δ* double mutant strains. (**H**) Quantification of difference in CPDs remaining (after 2hr repair) along the NTS between H3K36A *rad16Δ* double mutant and *rad16Δ* single mutant for genes associated with polyA SRAT (i.e., antisense transcripts) and those that are not (rest of genome).

## H3K36 methylation regulates canonical and cryptic TC-NER in yeast

Since the only known methylation target of Set2 is the H3K36 residue in histone H3, we hypothesized that mutation of H3K36 should not only result in similar UV sensitivity as *set2Δ*, but also similar effects on CPD repair. We confirmed that the H3K36A mutant had similar UV sensitivity as a *set2Δ* mutant analyzed in the same genetic background (histone deletion strain; S8A and S8B Fig) and resulted in a similar restoration of UV resistance in a GG-NER defective *rad16Δ* background (S8C Fig). Alkaline gel analysis (Fig 6A) indicated that H3K36A mutant did not significantly affect overall repair of CPD lesions compared to the histone WT control (Fig 6B); however, there was a marginal repair defect at the 3 hour (hr) time point (*P* = 0.107), similar to that observed in the *set2Δ* mutant.

CPD-seq analysis of repair at 2hr after UV irradiation revealed the expected pattern of faster repair along the TS relative to the NTS in the histone WT control (Figs 6C and S9A). Repair of

the TS was reduced in the H3K36A mutant (Figs 6D and S9B), as was repair asymmetry between the TS and NTS strand of yeast genes (Fig 6E). These results suggest that the H3K36A mutant causes a TC-NER defect, similar to *set2Δ*. Scaled CPD-seq analysis of the *rad16Δ* H3K36A double mutant relative to a matched *rad16Δ* control revealed faster repair of the NTS (Fig 6F), and a decrease in the repair asymmetry between the TS and NTS strands of yeast genes (Fig 6G). Repair of the NTS was especially prominent at SRAT-associated genes (Figs 6H, S10, and S11). These results confirm that Set2-catalyzed H3K36 methylation regulates repair of both TC-NER of the TS and cryptic TC-NER of the NTS of yeast genes.

## Discussion

While it has been established that histone modifications can promote repair of UV damage in chromatin by the GG-NER pathway, to what extent TC-NER is also regulated by histone modifications has been unclear. Here, we show that methylation of H3K36 by Set2 regulates TC-NER in yeast. Both UV sensitivity data and genome-wide repair studies indicate that Set2 methylation of H3K36 enhances repair of the TS of yeast genes throughout the genome. In contrast, Set2 suppresses cryptic TC-NER of the NTS by preventing antisense transcription in the coding regions of yeast genes, thereby establishing strand-specific repair asymmetry. For this reason, deletion of *SET2* or other genes that suppress cryptic antisense transcription (i.e., *EAF3*) partially restores UV resistance to yeast cells lacking GG-NER. Based on these data, we propose a model in which accumulation of H3K36 methylation behind a lesion-stalled RNA polymerase may serve as a signal to promote Rad26 recruitment and TC-NER along the TS, but this same modification acts to prevent cryptic transcription and TC-NER of the NTS (Fig 7). Because H3K36 methylation is often dysregulated in human cancers [51,52], these findings have potential implications for carcinogenesis.

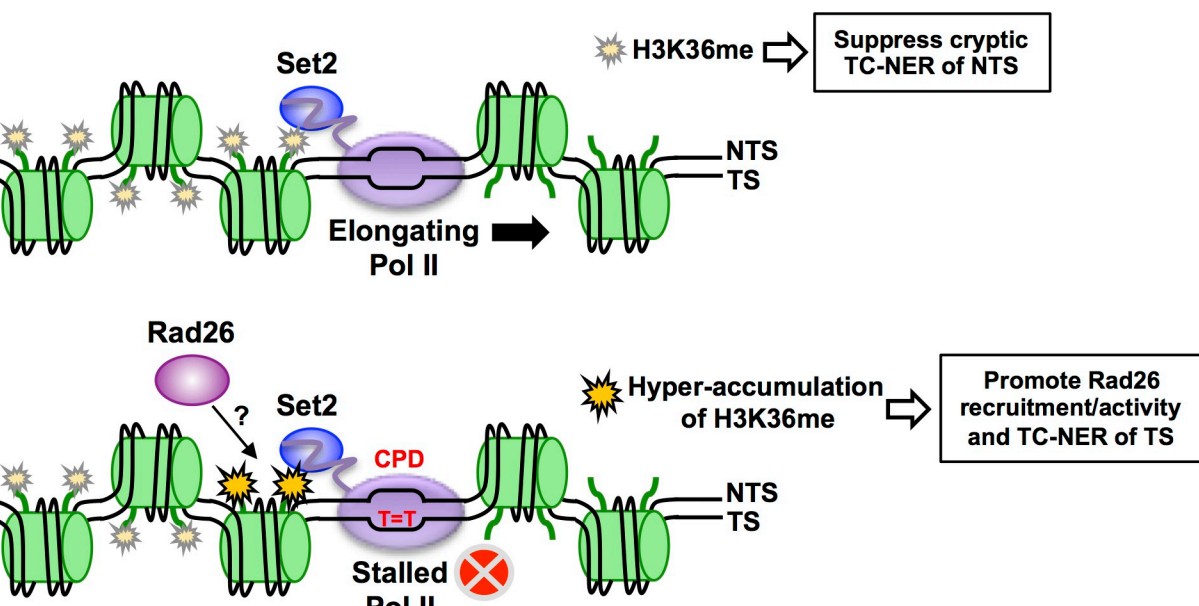

**Fig 7. Model of how Set2 methylation may promote TC-NER of the TS and suppress cryptic TC-NER of the NTS.** During normal Pol II elongation (top panel), co-transcriptional H3K36me by Set2 suppresses cryptic antisense transcription, thereby preventing cryptic TC-NER of the non-transcribed strand (NTS). When Pol II stalls at a CPD lesion (bottom panel), Set2-catalyzed H3K36me may accumulate immediately behind the stalled Pol II, thereby serving as an epigenetic signal of Pol II stalling. Hyper-accumulation of H3K36me may promote TC-NER of the transcribed strand (TS) by recruiting Rad26 behind the stalled Pol II.

Both UV sensitivity and genome-wide repair data indicate that Set2 facilitates efficient TC-NER of the TS of yeast genes. Deletion of *SET2* or mutation of H3K36 results in mild UV sensitivity, similar to that observed for mutants in the TC-NER pathway (e.g., *rad26Δ*), that is epistatic with Rad26. Furthermore, CPD-seq analysis indicates that repair of the TS is impaired in both *set2Δ* cells and GG-NER deficient *rad16Δset2Δ* cells. Similarly, the H3K36A mutation, which eliminates the Set2-catalyzed histone methylation site, causes a similar defect in TC-NER. While repair of the TS is slower in *set2Δ* mutant cells, loss of Set2 (or H3K36A) does not cause the same magnitude of repair defect as a *rad26Δ* mutant [40], suggesting Set2 functions to facilitate, but is not essential for, TC-NER. It has been recently reported that the Set2 homolog in *S. pombe* is also epistatic with NER factors in cells treated with methyl methanesulfonate (MMS), a DNA alkylating agent [53]. This observation can be explained by our finding that Set2 regulates TC-NER, since the TC-NER pathway may function to repair MMS-induced alkylation damage [54]. Set2-catalyzed H3K36 methylation may also regulate GG-NER in yeast, since we observe a marginal decrease in overall CPD repair in yeast by alkaline gel analysis and an overall decrease in repair in our normalized CPD-seq data. A role for H3K36me in facilitating both NER pathways could potentially explain previous reports that NER activity is elevated at exons in UV-irradiated human cells [55,56], since these genomic regions are enriched for H3K36me3 [57–59].

An important question is how Set2-catalyzed H3K36 methylation regulates TC-NER. One possibility is that loss of H3K36 methylation could affect TC-NER indirectly by altering transcription. While loss of H3K36 methylation is known to alter gene expression in yeast, it does not affect the expression of known NER genes, and generally causes an increase in transcription [50,60], which should stimulate TC-NER, not impair it. Moreover, our analysis indicates that Set2 regulates repair independently of Set2-dependent changes in gene expression (see Fig 2F). Alternatively, induction of antisense transcription of the NTS in the *set2Δ* mutant could interfere with TC-NER of the TS, perhaps by colliding with a lesion-stalled Pol II and preventing recruitment of TC-NER factors. However, we do not favor this model, since the TC-NER defect in *rad16Δset2Δ* cells is roughly similar between genes with Set2-repressed antisense transcription (SRAT genes) and those without antisense transcription (Figs 5A, 5B, and S6).

Instead, we propose that accumulation of H3K36 methylation may act as an epigenetic signal to promote TC-NER at a stalled Pol II. Set2 methylates histones during transcriptional elongation due to its association with Pol II [52], and the level of H3K36me3 deposition has been shown to depend on the dwell time of Pol II during elongation [61]. This suggests that the long dwell time associated with Pol II stalling at a CPD lesion should result in the local hyper-accumulation of H3K36me. Hyper-accumulation of H3K36me may function as an epigenetic signal for Pol II stalling and potentially promote Rad26 recruitment and/or activity during TC-NER (Fig 7). A previous study has suggested that H3K36 methylation may facilitate Rad26 recruitment, as Rad26 binding to the coding region of the *GAL1* gene is decreased in *set2Δ* mutant cells [25]. Accumulation of H3K36me behind a stalled Pol II may recruit Rad26 to initiate TC-NER. Moreover, Rad26 binds to the upstream ('back') side of Pol II to promote elongation past barriers and initiate TC-NER [62,63]. Accumulation of H3K36me upstream of Pol II during stalling may help properly orient Rad26 behind Pol II to perform these functions (Fig 7).

Alternatively, H3K36 methylation may also regulate Rad26 activity, since it has been reported that histone H3 and H4 tails are important for the activity of Rhp26 (*S. pombe* Rad26 homolog) in nucleosomes [64]. Future studies will be required to explore these possible mechanisms. One potential caveat to our model is that H3K36 methylation is not typically found near the TSS of yeast genes, yet *set2Δ* mutation affects TC-NER near the TSS (Fig 3). However, this apparent discrepancy can be explained by the prior observation that increased Pol II dwell

time (in this case due to slower Pol II elongation) can result in elevated H3K36me3 levels even near the TSS [61]. We propose that prolonged Pol II stalling at a CPD lesion near the TSS may cause a similar 'early' H3K36me accumulation adjacent to the stalled Pol II. Our data also indicate that Set2 is important for TC-NER on the TSS-proximal side of the +1 nucleosome, even though repair in this region is largely Rad26 independent [40,65]. It is possible that Set2 and H3K36 methylation may also regulate Rad26-independent TC-NER, perhaps through regulating the activity of Elf1 or Rpb9 [5,38].

Unexpectedly, our data also indicate that deletion of *SET2* (or mutation of H3K36) rescues the UV sensitivity of a *rad16Δ* strain, which is GG-NER deficient. Multiple lines of evidence suggest that *set2Δ* mutant compensates for the loss of repair activity along the NTS in GG-NER deficient cells by inducing antisense transcription so that the NTS is repaired by TC-NER (Figs 5D and 7). First, rescue of UV sensitivity by *set2Δ* requires Rad26, indicating that it is dependent upon TC-NER, and also Dot1, which is necessary for induction of cryptic transcription in *set2Δ* mutant cells [50]. Second, CPD-seq analysis of the *rad16Δset2Δ* double mutant indicates that there is more repair along the NTS of SRAT genes, in which cryptic antisense transcription is induced by deletion of *SET2*. However, the NTS is still repaired in *rad16Δset2Δ* mutant cells in non-SRAT genes. This may be due to low levels of cryptic antisense transcription in non-SRAT genes, since Set2 has been reported to regulate subsets of other classes of noncoding RNAs [46]. Alternatively, UV exposure may induce a more widespread induction of cryptic antisense transcription in *set2Δ* mutant cells, perhaps due to transient disruption of chromatin structure during NER. These findings suggest that activation of cryptic transcription may be an important mechanism to promote resistance to UV damage and other bulky DNA adducts, particularly since many differentiated mammalian cells have attenuated GG-NER activity [66].

Because histone H3K36 methylation is frequently dysregulated in human cancers, these findings have important implications for carcinogenesis. Set2 homologs, including NSD1, NSD2, NSD3, and SETD2, are frequently mutated, translocated, or amplified in a variety of human cancers [51,52], and oncohistone mutations that dysregulate H3K36 methylation cause pediatric brain tumors [67]. Moreover, amplifications or mutations in Set2 homologs in human cancers has been linked to chemotherapy resistance [68–70], including to chemotherapeutics that cause DNA lesions repaired by NER (e.g., cisplatin and melphalan). It will be important to determine to what extent H3K36 methylation regulates TC-NER in human cells, and whether this contributes to carcinogenesis and chemotherapeutic resistance in cancer.

## Materials and methods

### Yeast strains and plasmids

All strains were constructed in the BY4741 strain background. The gene deletions were constructed following published methods [71]. Histone mutant plasmids were constructed from pJW500 and introduced into yeast strain WY499 using plasmid shuffling, as described previously [36]. The *RAD14* intron deletion was done using the yeast CRISPR-Cas9 gene editing system as described [72] and was confirmed by Sanger sequencing.

### UV sensitivity assays

Yeast cells were grown in Yeast extract Peptone Dextrose (YPD) medium to mid-log phase. For spotting assays, cells were 10-fold serially diluted in fresh YPD medium and spotted on YPD plates. After exposing to different doses of UVC light (254 nm), plates were incubated at 30°C in the dark and images were taken after 3–5 days of incubation. For quantitative UV survival assay, diluted yeast cells were plated on YPD plates and exposed to the indicated UV

dose. The number of colonies on each plate was counted after incubating for 3 days at 30˚C in the dark and analyzed for three independent trials, each trial representing the average of three replicate experiments.

## CPD-seq library preparation and sequencing

Yeast cells were grown to mid-log phase, pelleted, re-suspended in dH$_2$O and irradiated with 125J/m$^2$ UVC light (254 nm). After UV treatment, cells were incubated in the dark in pre-warmed fresh YPD medium for repair. Cells were collected immediately after UV (0hr) and at different time points (2hr or 3hr) after UV irradiation and repair incubation at 30˚C. The 2hr time point was chosen since this time frame has been previously used to detect defects in TC-NER in yeast mutants (e.g., *rad26Δ* and *elf1Δ* [38,40]). The cells were pelleted and stored at the -80˚C freezer until genomic DNA isolation. Genomic DNA extraction, CPD-seq library preparation and quality control, sequencing with an Ion Proton sequencer, and data processing were performed as previously described [26,73]. The resulting sequencing reads were aligned to the yeast genome (saccer3) using bowtie2 [74]. Only CPD-seq reads associated with lesions at dipyrimidine sequences (i.e., TT, TC, CT, CC) were retained for further analysis. CPD-seq data for WT replicate 1 and *rad16Δ* strains were published previously [38,39].

Bin analysis for CPD repair along the transcribed strand (TS) and non-transcribed strand (NTS) of ~5000 yeast genes was performed as previously described [39], using transcription start site (TSS) and polyadenylation site (PAS, also referred to as transcription termination site (TTS) or transcription end site (TES)) coordinates from [39]. A similar gene bin analysis was used to display the fraction of CPDs remaining for yeast genes using the Java Treeview program [75,76], as previously described [41]. Genes were sorted by transcription rate [42] or by whether the gene is associated with a polyA SRAT. Transcriptional asymmetry analysis calculated the log$_2$ ratio of the fraction of unrepaired CPDs in the TS relative to the NTS for each bin, as previously described [41]. Single nucleotide resolution repair analysis adjacent to the TSS was performed for the ~5200 yeast genes with published TSS coordinates [37], as previously described [39,54]. Nucleosome dyad coverage from MNase-seq experiments was obtained from [77]. We used a high-resolution WT nucleosome positioning data set for comparison, since neither *set2Δ* nor *rad16Δ* causes a significant change in nucleosome positions in yeast [78,79]. CPD-seq data for WT and *set2Δ* mutant cells was scaled using the fraction of unrepaired CPDs measured by T4 endonuclease V alkaline gel assays for WT or *set2Δ* mutant cells (Fig 1G), as previously described [38]. Similar scaling was performed for the H3K36A and matched WT control CPD-seq data. CPD-seq data for the initial *rad16Δ* and *rad16Δset2Δ* experiments were not scaled, but the bin plot analysis of the replicate *rad16Δset2Δ* CPD-seq data was scaled (to remove variability due to differences in library preparation, etc.) so that the median ratio for all adjacent intergenic bins relative to the *rad16Δ* control was set to 1. Similar scaling was performed for the *rad16Δ*H3K36A bin plot analysis. Analysis of genes with Set2-repressed antisense transcripts (SRATs) used gene lists of either polyA SRATs or all SRAT-associated genes, published in [46].

## Western blot

Cells were cultured in YPD medium at 30˚C to late log phase and whole cell protein extracts were prepared from the cells using the procedure described previously [80]. The protein extracts were analyzed by western blot using antibodies against total H3 (ab46765, Abcam) and tri-methylated H3K36 (ab9050, Abcam). Blots were scanned using the Typhoon FLA 7000 (GE Healthcare).

### Analysis of bulk CPD repair in yeast

Yeast cells were grown in YPD media to mid-log phase, pelleted, resuspended in ddH$_2$O and irradiated with 100J/m$^2$ UVC (254nm). The UV-treated yeast cells were incubated in YPD media at 30°C with aliquots taken at the indicated timepoints and stored at -80°C until genomic DNA isolation. Genomic DNA was isolated using PCI (25:24:1) extraction, resuspended in TE pH 8, and RNase A treated for 30 minutes. Equal amounts of DNA were treated with T4 endonuclease V (courtesy of Dr. Steven Roberts, Washington State University) before resolving samples using alkaline gel electrophoresis. The resulting alkaline gels were then neutralized and stained using SYBR Gold (Invitrogen) before imaging with the Typhoon FLA 7000 (GE Healthcare). ImageQuant 5.2 was then used to measure the ensemble average pixel intensity of each lane (corrected by a no-enzyme control lane) and calculate the average number of CPDs per kilobase pair (kb). Percent repair for each timepoint was then calculated by normalizing the CPDs per kb for each timepoint to the 0 hour timepoint. Graphs represent the mean and SEM of three independent experiments.

## Supporting information

**S1 Fig.** (**A**) Western blot confirming that deletion of *SET2* or mutation of H3K36 eliminates H3K36 methylation. Western blot analysis using anti-H3K36me3 or anti-H3 (loading control) was used to probe total protein extracts isolated from the indicated yeast strains. Asterisk (*) indicates a nonspecific cross-reacting band recognized by the anti-H3K36me3 antibody. (**B**) UV sensitivity assay indicating that the intron in the *RAD14* gene is not required for *set2Δ* UV sensitivity. UV sensitivity assays were performed using the indicated UVC doses for the specified yeast strains. (TIF)

**S2 Fig. CPD-seq analysis of a time course of genome-wide repair of an independent experimental replicate of the *set2Δ* mutant. (A-D)** Bin plot analysis of the average fraction of CPDs remaining after 2 or 3 hours of repair was plotted from the CPD-seq data along the transcribed strand (TS) and non-transcribed strand (NTS) of ~5000 yeast genes in WT (**A,C**) and *set2Δ* (**B,D**) mutant cells. Each gene was divided into six equally sized bins and the fraction of unrepaired CPDs at the indicated time point relative to 0hr time point was plotted. Fraction of CPDs remaining in three bins (167 bp in length) upstream of transcription start site (TSS), as well as downstream of transcription end site (TES) was also plotted. (**E-F**) Log$_2$ ratio of unrepaired CPDs on the TS relative to the NTS in the WT and *set2Δ* cells at 2hr (**E**) or 3hr (**F**) repair time points. (**G-H**) Gene plot analysis of WT and *set2Δ* cells depicting the fraction of CPDs remaining following 2hr of repair (relative to 0hr) for both the TS and NTS. Genes are ordered based on their published transcription frequency [42]. (TIF)

**S3 Fig. Deletion of *SET2* in a *rad16Δ* GG-NER deficient background causes a defect in repair of the TS but promotes repair of the NTS. (A-B)** High-resolution analysis of the CPD-seq data showing the fraction of CPDs remaining after 2hr repair in *rad16Δ* (**A**) and *rad16Δset2Δ* (**B**) mutant cells around the TSS of ~5200 yeast genes. Average nucleosome coverage from [77] is shown for reference. (TIF)

**S4 Fig. Analysis of replicate *rad16Δset2Δ* CPD-seq data. (A)** Bin plot analysis of CPD-seq data for *rad16Δ* (data from [39]) and replicate *rad16Δset2Δ* experiment. Each gene was divided into six equally sized bins and the fraction of unrepaired CPDs at the indicated time point relative to 0hr time point was plotted. Fraction of CPDs remaining in three bins (167 bp in length)

upstream of transcription start site (TSS), as well as downstream of transcription end site (TES) was also plotted. CPD-seq data for *rad16Δset2Δ* mutant was scaled so that the median ratio of fraction of CPDs remaining for the *rad16Δset2Δ* relative to the *rad16Δ* control for the upstream and downstream intergenic DNA bins was set to 1. Transcribed strand (TS) is in red; non-transcribed strand (NTS) is in blue. (**B**) Quantification of repair asymmetry in replicate *rad16Δset2Δ* relative to the *rad16Δ* control, calculated as the log$_2$ ratio of unrepaired CPDs on the TS relative to the NTS for each bin.
(TIF)

**S5 Fig.** (**A**) Quantification of difference in CPDs remaining (after 2hr repair) along the NTS between *rad16Δset2Δ* double mutant and *rad16Δ* single mutant for genes associated with polyA SRAT (i.e., antisense transcripts) and those that are not (rest of genome). (**B**) Same as panel **A**, except for *rad16Δset2Δ* replicate experiment. Replicate CPD-seq data for *rad16Δset2Δ* mutant was scaled so that the overall median ratio of fraction of CPDs remaining for the upstream and downstream intergenic DNA bins was set to 1.
(TIF)

**S6 Fig.** (**A**) Bin plot analysis of CPD-seq data for *rad16Δset2Δ* double mutant and *rad16Δ* control (from [39]) for all genes associated with a Set2-repressed antisense transcript (all SRATs; not just polyA SRATs). Each gene was divided into six equally sized bins and the fraction of unrepaired CPDs following 2hr repair relative to 0hr time point was plotted. Fraction of CPDs remaining in three bins (167 bp in length) upstream of transcription start site (TSS), as well as downstream of transcription end site (TES) was also plotted. Transcribed strand (TS) is in red; non-transcribed strand (NTS) is in blue. SRAT gene list from [46]. (**B**) Same as panel **A**, except for genes not associated with an SRAT. (**C**) Quantification of difference in CPDs remaining between *rad16Δset2Δ* double mutant relative to the *rad16Δ* control along the NTS for each bin. Genes associated with an SRAT are depicted in purple; non-SRAT genes are shown in black.
(TIF)

**S7 Fig.** (**A-D**) UV sensitivity assays indicate that deleting *XRN1* or *RRP6* exonucleases to stabilize the SRAT transcripts does not modulate *set2Δ* UV sensitivity in WT background (**A,C**) or UV resistance in a *rad16Δ* mutant background (**B,D**). UV sensitivity assays were performed using the indicated UVC doses for the specified yeast strains. (**E,F**) UV sensitivity assays indicates that Ioc4 (subunit of the Isw1b complex) does not mimic *set2Δ* UV sensitivity (**E**) or modulate UV resistance in *rad16Δ* mutant background (**F**).
(TIF)

**S8 Fig. Deletion of *SET2* in histone strain results in similar UV sensitivity (or resistance) as H3K36A mutant.** (**A**) UV sensitivity assay showing sensitivity of *set2Δ* mutant (deleted in histone WT background) is similar to that of H3K36A mutant. UV sensitivity assays were performed using the specified UVC doses for the indicated yeast strains. (**B**) Graph of data for a quantitative UV sensitivity assay for the indicated yeast strains. Mean ± SEM is plotted for n = 3 replicates. Asterisks (*) indicate significant difference in UV sensitivity between WT and mutant strains ($P < 0.05$), using two-sided t-test. Source data is in S3 Data. (**C**) Same as panel **A**, except showing that UV resistance of *set2Δ* mutant (deleted in histone WT *rad16Δ* background) is similar to that of H3K36A mutant in a *rad16Δ* background.
(TIF)

**S9 Fig.** (**A-B**) High-resolution analysis of the CPD-seq data showing the fraction of CPDs remaining after 2hr repair relative to the 0hr time point in histone WT (**A**) and H3K36A (**B**)

mutant cells around the transcription start site (TSS) of ~5200 yeast genes. CPD-seq data was normalized using overall repair at 2hr from alkaline gel analysis of histone WT and H3K36A mutant. Average nucleosome coverage from [77] is shown for reference.
(TIF)

**S10 Fig. (A)** Bin plot analysis of CPD-seq data for *rad16Δ*H3K36A double mutant and *rad16Δ* control (histone WT) for all genes associated with a polyA Set2-repressed antisense transcript (SRAT). Each gene was divided into six equally sized bins and the fraction of unre-paired CPDs following 2hr repair relative to 0hr time point was plotted. Fraction of CPDs remaining in three bins (167 bp in length) upstream of transcription start site (TSS), as well as downstream of transcription end site (TES) was also plotted. Transcribed strand (TS) is in red; non-transcribed strand (NTS) is in blue. SRAT gene list from [46]. **(B)** Same as panel **A**, except for genes not associated with a polyA SRAT. CPD-seq data for *rad16Δ*H3K36A mutant was scaled so that the overall median ratio of fraction of CPDs remaining for the *rad16Δ*H3K36A mutant relative to the *rad16Δ* (histone WT) control for the upstream and downstream inter-genic DNA bins was set to 1.
(TIF)

**S11 Fig. (A)** Bin plot analysis of CPD-seq data for *rad16Δ*H3K36A double mutant and *rad16Δ* (histone WT) control for all genes associated with a Set2-repressed antisense transcript (SRAT; not just polyA SRATs). Each gene was divided into six equally sized bins and the frac-tion of unrepaired CPDs following 2hr repair relative to 0hr time point was plotted. Fraction of CPDs remaining in three bins (167 bp in length) upstream of transcription start site (TSS), as well as downstream of transcription end site (TES) was also plotted. Transcribed strand (TS) is in red; non-transcribed strand (NTS) is in blue. SRAT gene list from [46]. **(B)** Same as panel **A**, except for genes not associated with an SRAT. **(C)** Quantification of difference in CPDs remaining between *rad16Δ*H3K36A double mutant relative to the *rad16Δ* (histone WT) control along the NTS for each bin. Genes associated with an SRAT are depicted in purple; non-SRAT genes are shown in black. CPD-seq data for *rad16Δ*H3K36A mutant was scaled so that the overall median ratio of fraction of CPDs remaining for the *rad16Δ*H3K36A mutant relative to the *rad16Δ* (histone WT) control for the upstream and downstream intergenic DNA bins was set to 1.
(TIF)

**S1 Data. Source data for graphs of alkaline gel analysis shown in Figs 1G and 6B.**
(XLSX)

**S2 Data. Source data for graphs of quantitative UV sensitivity data shown in Fig 5G.**
(XLSX)

**S3 Data. Source data for graphs of quantitative UV sensitivity data shown in S8B Fig.**
(XLSX)

## Acknowledgments

We are grateful to Kaitlynne Bohm for constructing the *set2Δ* mutant yeast strain. We thank Weiwei Du and Mark Wildung for Ion Proton sequencing.

## Author Contributions

**Conceptualization:** Kathiresan Selvam, John J. Wyrick.

**Formal analysis:** Kathiresan Selvam, Dalton A. Plummer.

**Funding acquisition:** John J. Wyrick.

**Investigation:** Kathiresan Selvam, Dalton A. Plummer, Peng Mao.

**Methodology:** Peng Mao.

**Software:** John J. Wyrick.

**Supervision:** John J. Wyrick.

**Writing – original draft:** Kathiresan Selvam, Dalton A. Plummer, John J. Wyrick.

**Writing – review & editing:** Peng Mao.

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
