## [Decision Letter · Decision Letter 0]

30 Sep 2021

Dear John,

Thank you very much for submitting your Research Article entitled 'Set2 histone methyltransferase regulates transcription coupled-nucleotide excision repair in yeast' to PLOS Genetics.

The manuscript was fully evaluated at the editorial level and by 4 independent peer reviewers. The reviewers appreciated the work and find the story both important and interesting. However, they also raised substantial concerns about the current manuscript. Based on the reviews, we would be willing to re-review a much-revised version that address their major concerns.  We cannot, of course, promise publication at that time.

If you decide to revise the manuscript for further consideration at PLOS Genetics, please aim to resubmit within the next 60 days, unless it will take extra time to address the concerns of the reviewers, in which case we would appreciate an expected resubmission date by email to plosgenetics@plos.org.

[LINK]

We are sorry that we cannot be more positive about your manuscript at this stage. Please do not hesitate to contact us if you have any concerns or questions.

Yours sincerely,

Brian Strahl

Guest Editor

PLOS Genetics

Wendy Bickmore

Section Editor: Epigenetics

PLOS Genetics

Reviewer's Responses to Questions

**Comments to the Authors:**

Reviewer #1: The review is uploaded as an attachment.

Reviewer #2: In this manuscript the authors examine the role of the yeast Set2 enzyme that methylates histone H3 lysine 36 in transcription coupled nucleotide-excision repair (TC-NER) and global genomic-nucleotide excision repair (GG-NER). They demonstrate that Set2 has a role in global TC-NER since a set2 mutant is UV sensitive, it is epistatic to the TC-NER factor rad26 mutant in the repair of UV-induced DNA damage, and set2 mutants have higher levels of unrepaired DNA damage genome-wide compared to wild-type cells. They also provide evidence that loss of Set2-mediated H3K36 methylation enhances antisense transcription and partially suppresses UV sensitivity of a mutant defective in GG-NER because it enhances repair of the non-coding transcribed strand (NTS) over the transcribed strand (TS). They provide evidence that one mechanism by which Set2 loss partially suppresses GG-NER repair defects is due to impaired recruitment of the RPD3S deacetylase complex whose epigenetic reader subunit Eaf3 binds Set2-mediated H3K36 methylation. They also provide genetic evidence that the ability of a set2 mutation to suppress GG-NER depends on the Dot1 histone methyltransferase, which is required for antisense transcription in set2 mutants.

Overall, I found the manuscript to be well written and the work presented to be well-done and of potential interest to the readership of PLOS Genetics. However, I believe that while the data presented are of high-quality and provide new insight into the role that Set2-mediated H3K36 methylation plays in DNA repair through the TC-NER and GG-NER pathways, there are some important scientific deficiencies that still need to be addressed. Those scientific concerns are outlined below.

Scientific concerns

1) The authors argue that Set2 loss causes global defects in TC-NER but suppresses GG-NER defects due to the increase in antisense transcription normally repressed by Set2. Are the DNA repair defects in TC-NER solely due to transcriptional interference that causes impaired Pol II transcription? Or does the physical presence of the antisense transcripts prevent repair? Could the authors combine their set2, rad26, rad16, set2/rad26, and set2/rad16 mutants with overexpression of the nuclear exosome subunit Rrp6 and/or the Xrn1 exonuclease to determine if this enhances TC-NER and/or GG-NER repair. Additionally, what happens to UV sensitivity and DNA repair of set2 and/or rad26 mutants are combined with rrp6 and/or xrn1 mutants?

2) The authors provide data that indicates loss of Set2-repressed antisense transcription promotes partial suppression of the GG-NER defect in the rad16 mutant. They indicate that impaired recruitment specifically of RPD3S (through the Eaf3 subunit) functions in this process as it increases antisense transcription. However, Set2 mediated H3K36 methylation also recruits additional chromatin effectors to transcribed genes, including the ATP-dependent chromatin remodeling Isw1b complex via the Ioc4 subunit. This complex has an important role in nucleosome remodeling that could be important for repair through these pathways. The authors have not addressed whether such additional chromatin effectors that depend on Set2 activity also are contributing to the TC-NER and/or GG-NER pathways. They should provide at least some genetic evidence that loss of Set2 dependent recruitment of these additional chromatin effectors is not contributing to the TC-NER and/or GG-NER repair process.

Reviewer #3: Review is uploaded as attachment.

Reviewer #4: This study examines the role of Set2 histone methyltransferase in nucleotide excision repair using the elegant genome-wide CPD mapping method that they developed. They demonstrate that a Set2 mutation reduces transcription and leads to elevated antisense transcription with the overall consequence of reduced transcription-coupled repair in the transcribed strand and increased repair in the nominally non-transcribed strand. They also show that this aids in the survival of mutants defective in global repair (Rad16). While this is a plausible scenario, the data is not very compelling, and I have three specific concerns. First, the authors have not shown increased UV resistance or increased global repair of CPDs in Set2Rad16 double mutant verses the Rad16 single mutant (Fig.1). Second, the various genomic plots showing transcribed verses non-transcribed strands only show marginal differences, and such differences might be expected for a strain containing a mutation in a gene known to be involved in regulating several cell cycle and division genes (Cell Reports 20:2693). Finally, there is no convincing quantitative data measuring the levels of the presumed anti-sense transcription. Thus, I recommend that this manuscript is better suited for a journal with a more specialized readership.

**Have all data underlying the figures and results presented in the manuscript been provided?**

Reviewer #1: Yes

Reviewer #2: Yes

Reviewer #3: **No: **Figure 1DE as referred to in the text appears to be missing. It might be in 1FG, but that is discussed in the results as if it were a different experiment. As commented on in the review, this needs to be sorted out and fixed.

Reviewer #4: Yes

PLOS authors have the option to publish the peer review history of their article (what does this mean?). If published, this will include your full peer review and any attached files.

Reviewer #1: No

Reviewer #2: No

Reviewer #3: No

Reviewer #4: No

×

---

## [Decision Letter · Decision Letter 1]

8 Feb 2022

Dear Dr Wyrick,

We are pleased to inform you that your manuscript entitled "Set2 histone methyltransferase regulates transcription coupled-nucleotide excision repair in yeast" has been editorially accepted for publication in PLOS Genetics. Congratulations!

Yours sincerely,

Brian Strahl

Guest Editor

PLOS Genetics

Wendy Bickmore

Section Editor: Epigenetics

PLOS Genetics

Comments from the reviewers (if applicable):

Reviewer's Responses to Questions

**Comments to the Authors:**

Reviewer #1: I appreciate the authors' thoughtful response to the initial manuscript review. They thoroughly addressed all of my major concerns and provided detailed explanations for those issues that they did not address in the text.

Reviewer #2: I believe the authors have sufficiently addressed the concerns of the four reviewers in this revised manuscript.

Reviewer #3: Concerns have been appropriately addressed and the revised manuscript is much stronger than original submission.

**Have all data underlying the figures and results presented in the manuscript been provided?**

Reviewer #1: None

Reviewer #2: None

Reviewer #3: Yes

PLOS authors have the option to publish the peer review history of their article (what does this mean?). If published, this will include your full peer review and any attached files.

Reviewer #1: No

Reviewer #2: No

Reviewer #3: No

**Data Deposition**

http://datadryad.org/submit?journalID=pgenetics&manu=PGENETICS-D-21-01161R1

**Press Queries**

×

---

## [Editor Report · Acceptance letter]

4 Mar 2022

PGENETICS-D-21-01161R1 

Set2 histone methyltransferase regulates transcription coupled-nucleotide excision repair in yeast 

Dear Dr Wyrick, 

We are pleased to inform you that your manuscript entitled "Set2 histone methyltransferase regulates transcription coupled-nucleotide excision repair in yeast" has been formally accepted for publication in PLOS Genetics! Your manuscript is now with our production department and you will be notified of the publication date in due course.

With kind regards,

Zsofia Freund

PLOS Genetics

On behalf of:
